# Non-Conventional Risk Factors: “Fact” or “Fake” in Cardiovascular Disease Prevention?

**DOI:** 10.3390/biomedicines11092353

**Published:** 2023-08-23

**Authors:** Giovanni Cimmino, Francesco Natale, Roberta Alfieri, Luigi Cante, Simona Covino, Rosa Franzese, Mirella Limatola, Luigi Marotta, Riccardo Molinari, Noemi Mollo, Francesco S Loffredo, Paolo Golino

**Affiliations:** 1Department of Translational Medical Sciences, Section of Cardiology, University of Campania Luigi Vanvitelli, 80131 Naples, Italyfrancesco.loffredo@unicampania.it (F.S.L.);; 2Cardiology Unit, Azienda Ospedaliera Universitaria Luigi Vanvitelli, 80138 Naples, Italy; 3Vanvitelli Cardiology Unit, Monaldi Hospital, 80131 Naples, Italy

**Keywords:** cardiovascular diseases, conventional risk factors, cardiovascular prevention, emerging risk factors

## Abstract

Cardiovascular diseases (CVDs), such as arterial hypertension, myocardial infarction, stroke, heart failure, atrial fibrillation, etc., still represent the main cause of morbidity and mortality worldwide. They significantly modify the patients’ quality of life with a tremendous economic impact. It is well established that cardiovascular risk factors increase the probability of fatal and non-fatal cardiac events. These risk factors are classified into modifiable (smoking, arterial hypertension, hypercholesterolemia, low HDL cholesterol, diabetes, excessive alcohol consumption, high-fat and high-calorie diet, reduced physical activity) and non-modifiable (sex, age, family history, of previous cardiovascular disease). Hence, CVD prevention is based on early identification and management of modifiable risk factors whose impact on the CV outcome is now performed by the use of CV risk assessment models, such as the Framingham Risk Score, Pooled Cohort Equations, or the SCORE2. However, in recent years, emerging, non-traditional factors (metabolic and non-metabolic) seem to significantly affect this assessment. In this article, we aim at defining these emerging factors and describe the potential mechanisms by which they might contribute to the development of CVD.

## 1. Introduction

Despite tremendous advancements in prevention and treatment, CVDs are still the leading causes of mortality and the major contributors to disability in industrialized countries, with a huge impact on social and economic systems. Since the first observations from the Framingham Heart Study started in 1948 [1], several other epidemiological studies have confirmed the impact of the so-called conventional CV risk factors, such as age, blood pressure, glucose blood levels, lipid profile, and smoking status, as major determinants of CV disease development and clinical outcome [2]. Based on all these data, the current guidelines on cardiovascular prevention using the SCORE algorithm define the risk of fatal and non-fatal events in a 10-year period [3]. The achievement of targets for all the modifiable risk factors is the *primum movens* in prevention [3]. However, despite the major effort in promoting a healthy lifestyle and keeping the cardiovascular risk factors at target, in 2019, an estimated 17.9 million people died from CVDs, representing 32% of all global deaths. Of these deaths, 85% were related to heart attack and stroke [4,5,6,7]. Thus, the optimistic expectation of cardiologists to reduce the CVD burden because of improved prevention strategies and treatment of the modifiable risk factors has been largely unmet. Several aspects should be taken into account to explain the reasons of such failure. In December 2022, the American College of Cardiology (ACC) announced the publication of “The Global Burden of Cardiovascular Diseases and Risk: A Compass for Future Health”. In this document, 18 specific CV conditions and 15 risk factors across 21 global regions were analyzed to provide an up-to-date overview of the global burden of CVD [8]. This document includes data from 204 countries, analyzing the major global modifiable CVD risk factors, how they contribute to disease burden, and recent strategies for prevention [8]. Based on this analysis, hypertension, hypercholesterolemia, dietary lifestyle, and air pollution were the leading causes of CVD worldwide. A total of 15 leading risks for CV diseases were included and divided in three categories: environmental (air pollution, household air pollution, low and high temperature); metabolic (systolic blood pressure, low-density lipoprotein cholesterol, body mass index, fasting plasma glucose, kidney dysfunction); and behavioral (dietary, smoking, alcohol use, physical activity). This report has also evaluated the disability-adjusted life years (DALYs), looking at the years of life lost because of premature mortality, and years lived with disability [8]. As a main result of this analysis, ischemic heart disease remains the major cause of CV death, with up to 9.44 million deaths in 2021 and 185 million DALYs. Hypertension remains the modifiable risk factor mainly associated with premature CV deaths, with up to 10.8 million CV deaths and 11.3 million deaths overall in 2021 [8]. A dietary lifestyle evaluation has considered under-consumed food, such as vegetables, fruits, fiber, vegetables, and over-consumed food, such as meats, sodium, and sugar-sweetened beverages. This analysis reveals an association of 6.58 million CV deaths and 8 million deaths overall in 2021 [8]. However, the conventional risk factors evaluated in this latest document may explain only part of the cardiovascular disease burden. In the last few years, several epidemiological and experimental studies have linked the development of CVDs to novel and emerging risk factors [9], such as homocysteine and vitamin D levels, gut microbiota, sleep apnea, sleep duration, uric acid plasma concentration over the air pollution, and climate change, as already stated by the ACC document [8]. In the present manuscript, we will evaluate how these emerging non-conventional risk factors are linked to CVDs and how they should be managed for cardiovascular prevention.

## 2. Literature Sources and Search Strategy

We performed a non-systematic review of the literature by applying the search strategy in different electronic databases (MEDLINE, EMBASE, Cochrane Register of Controlled Trials, and Web of Science). Original reports, meta-analyses, and review articles in peer-reviewed journals up to June 2023 evaluating the clinical role of non-conventional risk factors in determining CVD in the general population. Homocysteine, uric acid, vitamin D, gut microbiota, sleep apnea, air pollution, global temperature, and sleep duration were incorporated into the electronic databases for the search strategy. The references of all identified articles were reviewed to look for additional papers of interest to extrapolate the more recent available data on the link between non-traditional risk factors and CVD.

## 3. Metabolic Risk Factors

### 3.1. Homocysteine: The Never-Ending Debate in Cardiovascular Prevention

Homocysteine is a sulphur amino acid that originates from the metabolism of methionine. Methionine, an essential food-derived amino acid, plays a vital role in cellular processes through the donation of methyl groups [10]. The first metabolite originating from methyl transfer is S-adenosyl methionine, which is subsequently converted to S-adenosyl homocysteine, the immediate precursor of homocysteine. The latter can be ‘recycled’ by taking the methylation route, resulting in the regeneration of methionine, or alternatively, it can be eliminated renally via the transulfuration route, leading to the formation of cysteine. Both processes are mediated by enzymes whose cofactors are vitamin B12, folic acid, and vitamin B6 [11,12]. Under physiological conditions, there is a balance between homocysteine formation and elimination [12]. If homocysteine can accumulate in the body, the biochemical transformation process fails, leading to a serum level increase [12]. Serum homocysteine values between 5 and 15 micromol/L are considered normal while mild hyperhomocysteinemia is defined as values between 15 and 30 micromol/L; moderate, between 30 and 100 micromol/L; and severe, if greater than 100 micromol/L [13]. In the healthy population, blood levels of homocysteine do not appear to be significantly influenced by dietary intake [14]. Hyperhomocysteinemia has many causes, with genetic profiles playing a dominant role: several genetic polymorphisms have been recognized [15] as responsible for the deficiency of enzymes involved in homocysteine metabolism [16]. The most frequent polymorphisms involve the gene coding for methylenetetrahydrofolate reductase and the one coding for cystathionine beta synthase [17]. Other causes include vitamin B12, B6 and folic acid deficiency [18]; advanced age; male sex; menopause; lifestyle habits, such as alcohol abuse and smoking [19]; and certain diseases, including cancers [15], chronic kidney disease [20], hypothyroidism [21], and inflammatory bowel disease [22]. Mention should be made of drugs that may interfere with the metabolism of homocysteine or its enzymatic cofactors: these include methotrexate, carbamazepine, nitrates, fibrates, and metformin [23].

Over the past few decades, the correlation of homocysteine with the incidence of cardio- and cerebrovascular events as well as its potential role in the pathogenesis of atherosclerosis have been the subject of countless debates [24,25,26]. The first correlation between serum homocysteine levels and the incidence of coronary artery disease is dated 1956 [27]. Numerous clinical studies and meta-analyses have subsequently supported this theory, reporting a 20% increase in the risk of new coronary events for every 5 micromol/L increase above normal serum homocysteine levels [28] and an increased risk of fatal and non-fatal coronary [29,30,31] and cerebrovascular events [30,32]. Further analyses corroborate these data, showing a 25% reduction in homocysteine levels (approximately 3 micromol/L) correlates with a lower risk of cardiac ischemic events and stroke [32].

The relationship between hyperhomocysteinemia and mortality for coronary artery diseases or cardiovascular causes or all causes has been evaluated in a meta-analysis of 20 prospective studies reporting that elevated homocysteine levels were an independent predictor of cardiovascular events, mortality from cardiovascular causes, and mortality from all causes [33].

Other studies have correlated hyperhomocysteinemia with an increased risk for and recurrence of venous thromboembolic events [34,35,36], peripheral artery diseases [37], and congestive heart failure [38].

Based on this evidence, hyperhomocysteinemia has been proposed as an independent cardiovascular risk factor [38,39].

Several cellular mechanisms have been proposed to explain how hyperhomocysteinemia is implicated in the etiology of cardio- and cerebrovascular events. Endothelial dysfunction, increased arterial stiffness, and a prothrombotic state are common in patients with hyperhomosysteinemia [40]. The main pathways associated with this endothelial impairment are: a) increased oxidative stress [41]; b) a reduction in the expression of the endothelial isoform of nitric oxide synthetase (eNOS) and increase in the cellular expression of caveolin-1 that is an inhibitor of eNOS, thus leading to a reduced release of nitric oxide [42]; and c) the upregulation of cell adhesion molecules, resulting in an increased chemotaxis of monocytes on the endothelium and increased endothelial expression of IL-8, which favor inflammatory processes [43].

Hyperhomocysteinemia is also associated to collagen synthesis [44] and vessel smooth muscle cell proliferation [45], through activation of cyclin A, protein kinase C, and the proto-oncogenes c-myc and c-fos [45,46] as well as increased production of phospholipids [46] and increased expression of platelet growth factor [47]. This smooth muscle cells proliferation as well as increased collagen deposition and alterations in elastic tissue composition [48] is responsible for increased arterial wall stiffness [49,50,51]. This phenomenon is facilitated by the inactivation of eNOS and the reduced production of nitric oxide [52]. A schematic view of homocysteine pathways involved in CVD is provided in Figure 1.

Moreover, several studies have also linked hyperhomocysteinemia to increased prothrombotic state [53]. This effect has been mainly related to: (a) factor XII and factor V activation [54]; (b) tissue factor expression [55]; (c) thrombomodulin inhibition [56] that results in a reduction of protein C activation [57]; (d) a reduction in the anticoagulant effect of antithrombin III, thus altering the binding capacity of endothelial heparan sulphate with the latter [58]; and (e) the reduction of plasminogen activator function and increased expression of its inhibitor [59].

In light of these basic findings, several clinical studies have investigated whether the treatment of hyperhomocysteinemia might result in cardiovascular benefits in terms of cardio- and cerebrovascular event reduction with conflicting results.

### 3.2. Uric Acid: Still a Controversial Cardiovascular Risk Factor?

Uric acid (UA) is the final product of purine metabolism. The increase in its blood levels may depend either on an increased production or on a reduced elimination [60]. If hyperuricemia develops, urate crystals accumulation may occurs in the joints leading to the clinical manifestations of gout, subsequently also affecting the renal parenchyma and the excretory tracts with the picture of gouty nephropathy and nephro/urolithiasis [61]. Beyond this known effect, several other clinical studies have also investigated the relationship between high blood levels of UA and the development of CVDs [62] and, as for homocysteine, with conflicting results. The Framingham Heart Study did not indicate hyperuricemia as an independent risk factor for coronary artery disease, cardiovascular death, and death from all causes [63,64]. Some epidemiological studies have described a J- or U-shaped relationship between UA levels and cardiovascular risk, meaning that patients with either very low or very high UA values have an increased cardiovascular risk [65]. More recently, clinical studies seem to support the role of hyperuricemia in atherosclerosis, systemic arterial hypertension, atrial fibrillation, and chronic kidney disease as the pathophysiological processes promoted by UA, such as oxidative stress and inflammation that are the basis of endothelial dysfunction, which may contribute to atherothrombotic events. An increase in the activity of the enzyme xanthine oxidase, which regulates the synthesis of UA and which uses molecular oxygen as an electron acceptor for its function, determines the formation of reactive oxygen species (ROS) [66]. ROS are responsible for the lipid oxidation and the reduction of the nitric oxide concentration, which causes the loss of the physiological vasodilating effect of the endothelium and determines a prothrombotic phenotype. UA also favors an increase in the deposition of low-density lipoproteins at the endothelial level and their uptake by macrophages, which are transformed into foam cells, thus starting the process of atherosclerosis [67]. More recently, it has been highlighted how endothelial cells (ECs) may acquire a prothrombotic phenotype by expressing functional tissue factor (TF) once exposed to increasing doses of UA that can be reversed by the preincubation with an uricosuric agent [68]. Moreover, the endothelial dysfunction induced by hyperuricemia also favors the expression on the cell surface of the adhesion molecules (CAMs) involved in the initiation of the atherosclerosis process. This mechanism appears to be regulated by a modulation of the NF-kappaB pathway, leading to the upregulation of TF on cell surface and downregulation of its natural inhibitor, the Tissue Factor Pathway inhibitor (TFPI) [69]. Furthermore, the inflammasome [70] seems also to be involved with an increase in caspase-1 function, which would promote a particular type of endothelial cell apoptosis, known as pyroptosis, and the release of TNF-alpha [71]. A summary of the possible mechanisms by which UA is involved in CVD is provided in Figure 2.

These basic findings have been corroborated by a more recent clinical evaluation on patients with acute coronary syndrome (ACS) [72] by reporting that higher UA levels are associated with higher C-reactive protein (CRP) and troponin values. Additionally, ACS patients with high UA levels showed an angiographic picture of multivessel coronary artery disease and complex atherosclerosis according to the Ellis classification [72]. As regards the relationship between hyperuricemia and systemic arterial hypertension, several studies have shown an increase in blood pressure in patients with increased uric acid. A meta-analysis that studied 55,607 patients showed that for each 1 mg/dL increase in uric acid, the incidence of arterial hypertension increases by approximately 13% [73]. At the basis of this relationship, there would be the lower release of nitric acid and the activation of the renin–angiotensin–aldosterone system promoted by uric acid, which determine vasoconstriction and consequent increase in blood pressure. A relationship between hyperuricemia and increased onset of atrial fibrillation (AF) has been highlighted by the ARIC study, which shows a 1.16-fold increase in the risk of AF in subjects, mostly female and of African origin, with high UA values [74]. Atrial remodeling induced by the inflammatory effects and oxidative stress related to UA seems to be the underlying mechanism [75]. In light of the relationship between hyperuricemia and increased cardiovascular risk, the current therapeutic options mainly are represented by allopurinol and febuxostat, which inhibit the enzyme xanthine oxidase, and therefore, the UA production could have a role in reducing the incidence of cardiovascular events.

### 3.3. Vitamin D: Light and Shadow in Cardiovascular Prevention

Vitamin D, commonly known as the “sunshine vitamin”, is an essential nutrient that plays a critical role in the absorption and regulation of calcium and phosphorus, essential minerals necessary for strong bones, teeth, and overall skeletal health [76]. Unlike other vitamins, the human body can produce vitamin D through exposure to sunlight [77]. The precursor form of vitamin D, indeed known as 7-dehydrocholesterol, is naturally present in the skin [78]. Upon exposure to UVB radiation emitted by sunlight, a photochemical reaction takes place, leading to the transformation of 7-dehydrocholesterol into pre-vitamin D3 [78]. Subsequently, through heat-induced isomerization, pre-vitamin D3 is converted into cholecalciferol, also known as vitamin D3. Another form of vitamin D, the Vitamin D2, also known as ergocalciferol, is primarily derived from plant-based sources and is commonly utilized in fortified food products and some dietary supplements. Vitamin D2 and D3 are fully activated through two consecutive hydroxylation reactions catalyzed by specific P450 isoenzymes. The First hydroxylation, which occurs on the carbon in position 25, takes place in the liver by vitamin D 25-hydroxylase (CYP2R1) to form the pro-hormone 25-hydroxyvitamin D. Due its solubility and BPD binding properties, the level of this metabolite better reflects the body’s vitamin D status. The second hydroxylation occurs on the carbon in position 1 by 25-hydroxyvitamin D-1alpha-hydroxylase renal (CYP27B1) and is responsible for the synthesis of the biologically active metabolite, 1,25-dihydroxyvitamin D [78].

Beyond its well-known role in bone health, vitamin D has garnered increasing attention in relation to cardiovascular health. Numerous observational studies have investigated the link between vitamin D levels and CVDs. Although the results show some degree of variability, they consistently highlight an inverse association between vitamin D status and the risk of developing CVD [79,80,81]. The inverse correlation between vitamin D status and CVD seems to be particularly strong in older adults [82,83]. Meta-analyses of epidemiological studies support the inverse correlation between vitamin D levels and CVD [82,84]. The correlation between vitamin D levels and arterial hypertension holds significant importance. Blood pressure tends to exhibit geographical and racial disparities, whereby the risk of hypertension tends to rise from south to north in the Northern hemisphere. A suggested explanation for this latitude-based correlation is that sunlight exposure may offer protection, potentially due to the influence of ultraviolet B (UVB) radiation or vitamin D [85]. This association appears to be supported by animal studies. Mice that lack the vitamin D receptor (VDR) or have a genetic deficiency in the 1-alpha-hydroxylase gene, which is responsible for vitamin D activation, have been shown to develop high renin hypertension and cardiac hypertrophy [86,87]. In vitro studies highlight a favorable cardioprotective effect of 1,25-dihydroxyvitamin D. It has been reported that the pretreatment of ECs with vitamin D reduce the expression and activity of TF and CAMs induced by oxidized lipids [68] or interleukin-6 [88], possibly preserving endothelial function.

All the putative cardiovascular mechanisms associated with vitamin D are provided in Figure 3.

While in vitro studies and epidemiological studies have provided promising insights into the potential cardioprotective effects of vitamin D, the results from randomized controlled trials (RCTs) in this field have been inconclusive to date. The majority of trials conducted so far have primarily focused on investigating the impact of vitamin D supplementation on bone health. In many cases, vitamin D supplementation has been administered alongside calcium supplementation. Meta-analyses of randomized controlled trials (RCTs) have demonstrated non-significant reductions in CVD events with vitamin D supplementation [89,90,91]. According to a Cochrane review, vitamin D supplementation was found to significantly reduce all-cause mortality when compared to a placebo or no intervention. However, the review did not demonstrate a significant impact on cardiovascular mortality [92].

### 3.4. Gut Microbiota: The Axis Heart–Intestine in CVDs Development

Gut microbiota is a community made up of 10^14^ microorganisms, in symbiosis with the host, with numerous functions, such as the fermentation of indigestible carbohydrates, synthesis of vitamin K and biotin, and promotion of mucosal immune system [93]. In recent years, emerging studies have considered gut microbiota as a “forgotten organ” with metabolic, endocrine, and immunological functions, relevant for human health [94]. The balance of microbiota, in terms of number and diversification of species present, depends on various factors: presence of modulators (antibiotics, probiotics, and prebiotics), host’s characteristics (genetic background, immune system, hormones), and environmental conditions (diet).

The imbalance of gut microbiota, defined intestinal dysbiosis, is involved in the pathogenesis of many diseases, including atherosclerotic CVDs [95].

A recent meta-analysis and systematic review [96] reported a decrease in Bacteroides and Lachnospira with an increase in Enterobacteria, Actinobacteria, and Verrucomicrobiota in patients affected by coronary artery disease (CAD).

Intestinal dysbiosis promotes atherosclerosis through various mechanisms: local infections with microbial translocation and systemic inflammatory state activation and the production of pro-atherogenic metabolites, acting on the cholesterol metabolism.

The formation of atherosclerotic plaque can be promoted by an infection of the arterial wall or a distant infection. Some studies report in the vascular wall the presence of DNA of the same bacteria found in human gut [97]. These data do not indicate that bacteria are a CAD etiological agent, but they suggest that these organisms can promote plaque formation or accelerate disease progression [98]. Moreover, even a distant infection can promote atherosclerosis. In fact, some bacteria can compromise the integrity of the intestinal barrier, favoring lipopolysaccharide (LPS) translocation to systemic circulation [99]. LPS interaction with Toll-like receptor 4 (TLR4) on immune cells’ surface activates the NF-kappaB pathway with the production of pro-inflammatory cytokines that alter tissue homeostasis [100]. In fact, the pro-inflammatory state increases insulin resistance, which favors the development of diabetes mellitus and obesity and determines macrophage infiltration in the vascular wall, which initiates atherogenesis [101]. Certainly, microbial translocation, secondary to altered permeability of intestinal barrier, determinates a sub-acute or chronic low-grade inflammatory state, which induces metabolic syndrome development. The use of probiotics and prebiotics has been evaluated as a tool to reduce the systemic inflammatory response through the modulation of gut microbiota [102].

Furthermore, gut microbiota has a metabolic activity greater than the host’s activity. Some microbial species are able to metabolize complex dietary carbohydrates, indigestible or partially digestible by humans, into short-chain fatty acids (SCFAs); Bacteroides are the principal producers of acetate and propionate, and Firmicutes are the principal producers of butyrate [103]. SCFAs may have anti-inflammatory effects [104]. Hence, the alteration of intestinal homeostasis correlates with systemic inflammation and, therefore, promotes atherogenesis [95]. Moreover, some substances (choline, carnitine, betaine), contained in some nutrients, such as red meat, are metabolized by gut microbiota into trimethylamine, subsequently oxidized by hepatic flavin monooxygenase (FMO) into trimethylamine N-oxide (TMAO) [105], a pro-atherogenic metabolite. At a systemic level, TMAO causes endothelial dysfunction, a crucial phase in the pathogenesis of atherosclerosis, and increases platelet calcium signaling with a pro-thrombotic effect [106,107]. TMAO blood levels are proportional to atherosclerotic plaque vulnerability and evaluated with optical coherence tomography (OCT). These data confirm TMAO’s pathogenetic role in atherogenesis but also suggests its potential role as a biomarker of coronary plaque progression [108].

In addition, gut microbiota has important effects on cholesterol metabolism. In fact, there are bacteria that metabolize primary bile acids, produced in the liver from cholesterol, into secondary bile acids. These, through farnesoid X receptor (FXR) and G protein-coupled TGR5 receptor, have effects on the host’s metabolic activity (hepatic accumulation of triglycerides) and on the inflammatory state [109]. Alterations in gut microbiota influence the type of secondary bile acids that are produced. Changes in the typology of bile acids correlate with metabolic disturbances. For example, an increase in 12α-hydroxylated bile acids (cholic acid, deoxycholic acid) correlate with insulin resistance development [110]. Recently, emerging data correlate blood cholesterol levels with different microbial species. In particular, Bacteroides reduce blood cholesterol levels through various mechanisms, mainly via the esterification of cholesterol into coprostanol that is not absorbed in the intestine and, therefore, eliminated with faeces and the inhibition of cholesterol synthesis [111].

Finally, a relationship between gut and thrombus microbiota in patients presenting with ACS has been also reported [112] with Prevotella coronary thrombus content remarkably increased and associated with higher thrombus burden, TMAO, CDL40, and vWF, especially in hyperglycemic ACS patients [112]. These data support the role of TMAO in increasing coagulation.

All the possible atherosclerotic mechanisms associated with gut microbiota are summarized in Figure 4.

Despite this growing evidence, the relationship between gut microbiota and CVDs is still under intensive investigation.

### 3.5. Lipoprotein(a): Unveiling the Enigmatic Lipid Particle

Lipoprotein(a), often abbreviated as Lp(a), is a lipoprotein particle that has garnered significant attention in the field of atherosclerotic CVD, becoming subject of intense research and debate in the last few decades [113]. The Lp(a) structure consists of a large and highly polymorphic glycoprotein referred to as apo(a) covalently bound to a molecule of apoB-100 [114]. In normotriglyceridemic individuals, apo(a) primarily associates with low-density lipoproteins (LDL). However, in dyslipidemic patients, apo(a) can also combine with apoB100 found in triglyceride-rich particles, specifically very low-density lipoproteins (VLDL) and intermediate-density lipoproteins (IDL) [115]. By the biochemical point of view, apo(a) is characterized by loop-like structures known as kringles, a structural motif also found in other coagulation factors, such as plasminogen (PLG), prothrombin, urokinase, and tissue-type PLG activators [114]. Elevated Lp(a) levels are thought to significantly contribute to atherosclerosis, primarily by interfering with macrophages [116]. Specifically, the macrophage’s receptor for VLDL can engage with a high affinity to Lp(a), facilitating its breakdown via endocytosis within lysosomes, resulting in its degradation and prompting the formation of foam cells with the deposition of cholesterol in atherosclerotic plaques [116]. This hypothesis gains support from observations that Lp(a) is widely present in human coronary atheroma and is more abundant in tissue from culprit lesions of patients with unstable coronary disease when compared to those with stable disease [117]. Furthermore, oxidized phospholipids present on Lp(a) trigger inflammation through a TLR 2-mediated pathway, exacerbating endothelial disfunction and contributing to increased inflammation within the arterial wall [118]. Lp(a) may also affect the coagulative homeostasis enhancing TF-mediated thrombosis and restrain the dissolution of clots [119], interfering with fibrinolysis competing with plasminogen [120,121]. However, therapeutic efforts to reduce Lp(a) levels using an mRNA inhibitor (Pelacarsen) did not result in changes to fibrinolysis, suggesting that negatively affecting fibrinolysis might not be a clinically significant characteristic of Lp(a) [122]. Large epidemiological studies support a strong correlation between Lp(a) levels and atherosclerotic CVD [113]. Pooled data derived from 36 prospective studies involving a total of 126,634 participants revealed that age and sex corrected risk ratio for CVD increases with each rise in standardized concentrations of Lp(a) [123]; elevated CVD risk persisted even after adjusting for conventional CV risk factors [123]. Lp(a) concentration shows consistent associations also with risk of stroke [123]. Recently, Lp(a) has been linked also to the inflammatory and calcification processes that underlie aortic valve degeneration and progression of aortic stenosis [124]. A summary of its putative mechanisms is provided in Figure 5.

### 3.6. The Metabolic Syndrome: A Cocktail of Ingredients Interconnected with Cardiovascular Risk

Metabolic syndrome (MS) is defined as the presence of at least three diagnostic criteria (central obesity, hyperglycemia, HTN, hypertriglyceridemia, low high-density lipoprotein (HDL)) [125]. Its correlation with increased CV risk has been well characterized [125]. A multifactorial pathogenesis underlines this condition with inflammation and insulin resistance (IR) as key playmakers [125,126]. IR, characterized by a reduced cellular response to insulin, determines MS development through various pathways [127]. It is well established that IR is linked to obesity through several mechanisms (the alteration of glucose transport by down-regulation of GLUT4 and increased expression of protein tyrosine phosphatases, which dephosphorylate and interrupt intracellular signaling) [128]. Furthermore, hyperinsulinemia, secondary to IR, is also responsible for obesity [128]. IR determines development of HTN due to reduced NO production by ECs [129] and hyperactivation of the sympathetic system [129]. Lipid metabolism alterations are also induced by IR [130]. In particular, the increased release of fatty acids from adipocytes causes increased hepatic VLDL secretion and, therefore, hypertriglyceridemia. VLDL stimulates the exchange of cholesterol esters from HDL, reducing its bioavailability for reverse cholesterol transport [130]. A schematic view is provided in Figure 6.

Strict glycemic control has a cardioprotective action through anti-inflammatory, anti-oxidative mechanisms with a reduction in endothelial dysfunction [131,132]. However, despite the achievement of glycemic compensation, CVDs continue to develop. The improvement of insulin sensitivity, through drugs such as metformin, leads to a reduction in cardiovascular events [133]. This suggests that CV risk is more related to IR than to blood glucose levels [134]. Thus, a marker of IR should considered by the current guidelines to better evaluate CV risk [134]. On this regard, HOMA index is a well-established marker of IR [135] with a defined prognostic value in CV patients [136] and it should be add to the current score for CV risk estimation.

A summary of mechanisms involved in the relationship between non-metabolic risk factor and CVDs has provided in Table 1.

## 4. Non-Metabolic Risk Factors and Surrogates

### 4.1. Obstructive Sleep Apnea Syndrome: The Diving Board to CVDs

Obstructive sleep apnea (OSA) syndrome is a clinical condition characterized by cyclical episodes of total (apnea) or partial (hypopnea) collapse of the upper airways, occurring during sleep, with the persistence of thoracoabdominal movements. At the end of the events, arousal occurs with transient hypoxemia, autonomic alterations, and sleep fragmentation [137].

Apnea is defined as a reduction in airflow of at least 90% compared to the basal one, lasting at least 10 s while hypopnea is defined as a reduction in airflow of at least 30%, for no less than 10 s, associated with a reduction of at least 3% in oxygen saturation (SaO_2_) [138].

The severity of OSA is based on the number of events/hour, and it is defined as AHI index (apnea/hypopnea index). Specifically, <5 events/hour define a normal respiratory pattern, 5–14 events/hour a mild apnea, 15–29 events/hour a moderate apnea, and from 30 events/h a severe apnea [138]. The gold standard for the diagnosis of OSA is represented by polysomnography (PSG) [138].

A diagnosis of OSA is made based on nocturnal breathing disorders (snoring, breathing pauses in sleep, restless sleep, awakening choking) and/or daytime sleepiness symptoms associated with an AHI > 5; on the contrary, if the AHI index is greater than 15, OSA can be diagnosed in the absence of symptoms [139].

In general population, OSA prevalence is approximately 34% in men and 17% in women [137,140] while in CVD populations, it ranges from 40% to 60% [141,142].

During sleep, a failure of the neuromuscular reflex that preserves the patency of the airways occurs, resulting in hypoxemia and hypercapnia, determining an increase in the respiratory effort and an awakening of a few seconds, which restores patency of the upper airways, thanks to a series of reflex mechanisms. When sleep resumes, the cycle repeats [143].

OSA represents an independent risk factor for CVDs, such as HTN, AF and other arrhythmias, HF, CAD, stroke, pulmonary hypertension, metabolic syndrome, and diabetes as shown in Figure 7. The involved mechanisms are multiple and probably interconnected.

During the apneic phase, by stimulating peripheral and central chemoreceptors [144], hypoxia and hypercapnia determine the activation of the sympathetic nervous system with consequent peripheral vasoconstriction and an increase in vascular resistance and heart rate [145]. This results in an increase in left ventricular afterload and cardiac work. In addition, there is an overall increase in left ventricular transmural pressure (that is the difference between ventricular systolic pressure and intrathoracic pressure) with increased wall stress [146,147]. The cycle repeats many times every night; therefore, the cardiovascular system is chronically exposed to neuro-hormonal stress, and the hyperactivity of the autonomic nervous system also extends to the daytime hours over time [145,148].

Intermittent hypoxia is also responsible for an increase in oxidative stress [149]: during the hypoxic phase, the cells adapt to an environment with low oxygen content, and with the reoxygenation phase, there is a sudden increase of oxygen with ROS formation, leading to cellular damage in the ischemic tissue [150,151].

Furthermore, a reduction in the levels of circulating NO has also been highlighted during OSA [152], and this could be implicated in endothelial dysfunction [153].

OSA is present in up to 30–50% of HTN patients, and 80% of patients with resistant HTN have OSA [139,154], representing an independent risk factor [137]. In patients with OSA, due to the overactivity of the sympathetic nervous system, the physiological reduction in blood pressure during the night (which configures the “dipper” profile) does not occur [155,156]. Therefore, there seems to be a correlation between sleep apnea and the non-dipper profile of essential HTN [157,158]. Furthermore, several randomized trials and meta-analysis have shown a reduction in blood pressure in patients with sleep apnea treated with CPAP [137,159].

OSA is associated with heart rhythm disturbances and sudden death; pauses and bradycardia are common in patients with OSA [139].

OSA is also an independent risk factor for AF with several pathophysiological mechanisms implicated. In particular, sudden changes in intrathoracic pressure can cause atrial remodeling and atrial fibrosis with consequent electrophysiological alterations [160]. Moreover, the sudden increase in sympathetic activity during apneas can lead to the activation of catecholamine-sensitive atrial on channels, thus determining focal discharges from which AF can be originated [161]. OSA is also associated with an increase in systemic inflammation, which may contribute to the genesis of AF [162].

Sleep apnea also increases the risk of CAD by favoring atherosclerotic process via oxidative stress, endothelial dysfunction, inflammatory state, and autonomic dysfunction. It has been reported that in OSA patients, myocardial infarction occurs more frequently during the night hours [163], and a higher pro-inflammatory profile is present [164] with an effective reduction of the latter if CPAP therapy is used [164]. This study, therefore, suggests that OSA could activate vascular inflammation with non-traditional pathogenetic mechanisms.

OSA is also a risk factor for incident strokes, stroke recurrence [165], and functional and cognitive outcomes [166].

Pulmonary hypertension is closely related to OSA. Hypoxia and hypercapnia induce arteriolar vasoconstriction in the short term and vascular remodeling in the long term that could lead to an irreversible increase in pulmonary vascular resistance and the development of pulmonary hypertension [167].

Sleep apnea, mainly the central form (CSA), is highly prevalent in HF patients as well, ranging from 40% to 60% of symptomatic patients [168].

OSA is also linked to obesity and metabolic syndrome since chronic intermittent hypoxemia and sleep loss is associated to higher plasma leptin levels [169], glucose metabolism impairment, and insulin resistance [170].

At least, there is a reciprocal interaction between obesity and OSA where they both reinforce their progression and their severity in a vicious circle. It is believed that the deposition of fat in the upper airways and the functional alteration of the airways themselves are the mechanisms involved in the pathogenesis of OSA in the obese subjects [171]. On the other hand, daytime sleepiness and decreased physical activity together with hyperleptinemia are the mechanisms probably implicated in weight gain in OSA.

### 4.2. Air Pollution: Health Breath as Part of Prevention

Air pollution is the contamination of the environment, indoor or outdoor, by a mixture of chemical, physical, or biological agents that change the characteristics of the atmosphere and even at low concentrations cause damage to human health, other living organisms and the environment [172]. According to the Global Burden of Disease (GBD) report, air pollution was responsible for 6.7 million deaths in 2019 alone [172,173]. Globally, nearly 20% of CVD deaths are attributable to air pollution [173]. The main components of this mixture of pollutants are Total Suspended Particulate Matter (PM), gaseous compounds including ozone (O_3_), nitrogen dioxide (NO_2_), carbon monoxide (CO), sulfur dioxide (SO_2_), and volatile organic compounds including benzene [172]. According to the World Health Organization, 99% of the world’s population breathes air that contains annual average levels of air pollutants that exceed guideline recommendations. Particularly high exposures have been documented in cities in Asia, western sub-Saharan Africa, and Latin America [172]. The most consistent evidence on health damage is attributed to PM, i.e., the set of airborne particles, ranging in diameter from 0.1 to 100 mm, capable of remaining in suspension in the air even for long periods [174,175]. Short- and long-term exposure to PM is associated with increased morbidity and mortality, impacting the progression of atherosclerosis [176], ischemic heart disease [177,178,179], stroke [180], and lung disease as well as the course of pregnancy and the health of newborns [181]. PM_10_ (particles between 2.5 and 10 mm in diameter) and, largely, PM_2.5_ (diameter < 2.5 mm), are the most linked to CVD and affecting global public health [182,183]. Lung inflammation and oxidative stress pathway is the primary response to air pollution exposure [184], contributing to the development of a systemic pro-inflammatory state and activation of secondary effector pathways that result in endothelial dysfunction, increased atherosclerotic plaque vulnerability, and the activation of a prothrombotic and proarrhythmic state [177,185,186]. Experimental animal models seem indeed to support this hypothesis [187]. Moreover, human exposure to pollutant nanoparticles causes their translocation into the systemic circulation through the alveolus-capillary membrane, interacting with the endothelium, accumulating at sites of vascular inflammation, thus favoring atherosclerotic process [188,189,190], with effects similar to those observed in the lungs [191] and thrombotic complications [192]. A relevant change in platelet function toward increased prothrombotic tendency has been confirmed in diabetic patients after recent (within two hours) exposure to PM [193]. In addition to these mechanisms, short-term PM_2.5_ exposure in animal models is associated with sympathetic nervous system activation and hypertension, probably mediated by neuroinflammation [194,195]. In a meta-analysis of 33 studies, short-term exposure to PM_2.5_ was associated with a significant decrease in heart rate variability (HRV) [196]. Decreased HRV is an index of autonomic system dysfunction and predicts an increased risk of cardiovascular morbidity and mortality in patients with heart disease [197]. Increased blood pressure and decreased HRV suggest an autonomic imbalance in favor of sympathetic tone and could further explain the rapid cardiovascular responses associated with air pollution, such as the initiation of fatal tachyarrhythmias and increased myocardial infarctions [177], as confirmed by the available literature [198]. High short-term exposure to PM_2.5_ is associated with an increased risk of acute coronary event, acutely destabilizing and rupturing atherosclerotic plaque, in patients with clinically significant pre-existing CAD but not in those with uninjured coronary arteries [199]. Moreover, short-term exposure to elevated levels of PM_2.5_ and PM_10_ is also associated with increased daily hospitalizations for STEMI and increased incidence of STEMI-related ventricular arrhythmias and cardiac death [200]. The effect of long-term exposure to major air pollutants was assessed by the ESCAPE study that have evaluated the incidence of acute coronary events in 11 European cohorts. At a mean follow-up of 11.5 years, exposure to annual mean levels of PM_2.5_ > 5 μg/m^3^ and PM_10_> 10 μg/m^3^ was associated with a 13% and 12% increase in the risk of nonfatal acute coronary events, respectively, with no evidence of heterogeneity between cohorts [201]. Other observational studies and meta-analyses have reported a positive correlation between long-term exposure to air pollution and the development and progression of subclinical atherosclerosis and calcium accumulation [202] as well as increased carotid intima-media thickness [203]. Based on the published data, no more doubts should exist on the role of air pollutants in CVD development. A schematic view of the relationship between air pollution and CVD is provided in Figure 8.

### 4.3. Climate Change: The Impact of Temperature

Temperature and its extreme variation is now recognized as a cardiovascular risk factor [204,205,206]. A very recent analysis evaluating 32,000 cardiovascular deaths in 27 countries on 5 continents over 40 years support the role of extremely hot or cold temperatures in determining heart disease deaths [206]. Mortality and morbidity induced by climate change are not exclusively due to hypothermia or hyperthermia, but also to indirect causes, such as respiratory diseases and CVDs, which can be undetected when the human body tries to adapt to climate changes [207]. A relationship between mortality from CVD and temperature exists with a U-, V-, or J shaped curve [208,209,210]. While the correlation between temperature and CVD has been established, the role of diurnal temperature range (DTR), defined as the difference between the maximum and minimum temperatures recorded in one day, in determining CV events needs to be better evaluated. Extreme cold weather conditions associated to climate change contributes to an increase in temperature variability that might increase clinical cardiovascular events [205]. It is known that exposure to cold activates both the sympathetic nervous system (SNS) and the renin-angiotensin-aldosterone system (RAAS), which interact with each other, leading to HTN and myocardial damage [211]. Skin blood flow (SBF) is reduced in response to cold due to vasoconstriction and increased urine output, thus inducing dehydration, hemoconcentration, and hyperviscosity [212]. Furthermore, eNOS and adiponectin inhibition contributes to endothelial dysfunction and lipid deposition, thus favoring atherosclerosis and plaque instability. Cold exposure also triggers mitochondrial dysfunction with myocardial damage, cardiac hypertrophy, and cardiac dysfunction. The increase in cardiac work and peripheral resistance contributes to an increase in oxygen consumption and a reduction in the ischemic threshold [211], which is clinically relevant, especially when the coronary circulation is already compromised.

On the contrary, exposure to heat leads to increased blood flow and sweating with loss of fluids and dehydration. The resulting hemoconcentration and hyperviscosity may cause thromboembolism, leading to increased risk of ischemic stroke [213]. In the presence of heat stroke, the increase in core temperature redistributes the flow on the skin to facilitate heat loss. Intestinal blood flow is reduced, and this could cause increased permeability of the intestinal epithelium, allowing bacteria, their toxic cell wall component LPS, or HMBG1 to move from the intestinal lumen into the circulation. TLR4 recognizes these molecules, stimulating innate and adaptive immune responses and causing systemic inflammatory response syndrome (SIRS). Along with this, hyperthermia induces the occlusion of arterioles and capillaries (microcirculatory thrombosis) or excessive bleeding (consumptive coagulation), leading to multiorgan dysfunction. The putative mechanisms linking climate changes and CVD is provided in Figure 9.

-Gender: historically, sex differences in thermoregulation were often assumed due to anthropometric factors. However, there is no evidence that women are at greater risk of heat illness when the usual risk-management techniques are in place regarding exercise intensity, clothing, and hydration [214]. It is still matter of debate whether the documented influences of reproductive hormones on thermoregulatory mechanisms in women result in quantifiable differences between the sexes in the capacity to dissipate heat [214]. In males, winter cold may play a role in the constriction of major epicardial vessels. In women, the greatest number of events occurs in the autumn and not in the winter, of which the mechanism remains unclear and should consider the different coronary anatomy (less elastic, smaller coronaries and fewer collateral circulations) [215]. In women in whom microvascular angina is more common, cold exposure could exacerbate its onset [216]. Furthermore, women have a higher temperature threshold beyond which the sweating mechanisms are activated and a lower production of sweat than men, which leads to less heat loss by evaporation and greater susceptibility to the effects of heat. Conversely, males had a greater reduction in core body temperature when exposed to cold, which could explain the higher cardiovascular risk and mortality in response to the cold [214]. Despite these pathophysiological difference, a recent meta-analysis indicates that gender did not affect the seasonal dynamics of myocardial infarction, with a trend of higher susceptibility in men than in women [217].-Age: the elderly are more vulnerable to low temperatures, whose thermoregulatory capacity is often compromised (especially 65–75 or >75 years) [216,218], with exposure to heat, people > 60 years respond with less sweating, reduced blood flow to the skin, less increase in cardiac output, and less redistribution of splanchnic and renal blood flow than younger people. On the other hand, during exposure to the cold, elderly people respond with reduced peripheral vasoconstriction (implying greater heat loss) and reduced metabolic heat production.-Regional differences: people living in metropolitan areas have greater socio-economic resources, medical resources, and a better ability to adapt, with lower mortality than people living in rural areas [219].-Occupational exposure: heat exposure is an increasingly severe challenge, especially to those susceptible occupations (miners, farmers) [220].-Diabetes: characterized by endothelial dysfunction and hypercoagulability. Several factors, such as oxidative stress and protein kinase C, could contribute to microvascular damage from hyperglycemia. The cold could affect diabetic patients more. The impaired thermoregulation and the reduced autonomic control could explain why diabetic patients are more vulnerable to warm temperatures [221].-Cardiovascular diseases: patients with prior MI are more susceptible to extreme temperatures; endothelin 1, an indicator of vascular damage, is higher in these patients in response to cold than in the healthy population.-Kidney disease: renal disorders are commonly associated with increased blood pressure, which is also an additional effect of extreme cold temperatures.-Hypertension: among patients with a history of hypertension, increased urea/creatinine levels, a marker of dehydration, have been observed in response to climate change.

Traditional risk factors as well as hormones and environmental factors (air pollution and infections) have seasonal variability with a winter cluster [222,223].

A negative relationship has been also observed between cardiovascular events and humidity [224]. When the air has a high percentage of humidity, perspiration and thermal homeostasis processes could be impaired, which would increase respiratory fatigue and heart rate [224].

In recent years, the increased concentrations of greenhouse gases due to human activities have led to an increase in temperatures. Unfortunately, the modification of this risk factor requires a major effort worldwide with green political strategies able to reduce the impact of global warming in the next few decades.

### 4.4. Sleep Duration: Is There a Right Time for Cardiovascular Benefits?

The correlation between sleep duration (even napping) and CVD has been investigated in the last few decades. Some studies focused on “short sleep”, defined as sleep time < 6 h/night, while others have focused on “long sleep”, defined as sleep time > 9 h/night [225]. The most dated studies do not support this correlation [226]. However, recent evidence suggest a link between sleep duration and CVD development and outcome [227,228,229,230].

The MORGEN study (Sleep Duration and Sleep Quality in Relation to 12-Year Cardiovascular Disease Incidence) [231] has evaluated sleep length and quality in 20,432 subjects between 20–65-year-olds with no previous diagnosis of CVD during a follow-up period of 10–15 years. The population was stratified into short sleepers (<6 h), normal sleepers (7–8 h), and long sleepers (>9 h). Short sleepers showed a 15% higher risk of CVDs and 23% higher of CHD that increased up to 63% and 79% if a short sleep duration was associated with poor sleep quality. According to these data, a long sleep duration was not associated with increased risk of CVD or CHD. It has been reported that sleep restriction is associated with metabolic changes [232] with impaired fasting glucose (probably because of elevation in cortisol level) and higher energy intake due to altered production of hormones, such as leptin and ghrelin [233]. In addition, hyperactivation of the sympathetic branch of the autonomic nervous system, inflammation pathways (including secretion of IL-6 and TNF-alpha), and oxidative system proteins (such as myeloperoxidase) have been described [234]. Moreover, a higher risk of HTN and metabolic syndrome as well as higher arrhythmic risk (mainly AF) have also been linked to sleep deprivation [235,236,237]. More recent evidence has led researchers to reconsider the correlation between prolonged sleep duration (>9 h) and cardiovascular risk, such as stroke, CVD, CHD, obesity, and diabetes mellitus [238]. This risk is exponentially related with an increase in the hours of sleep. The PURE study, enrolling 116,632 subjects from seven different regions, showed a J-shaped correlation between sleep hours and mortality or major cardiovascular events, with an estimated minimum risk between 6–8 h/day of sleep, including both night and daytime rest (daytime naps) [230]. These findings were corroborated by other observations, too [239,240,241]. A more recent prospective study on 33,883 adults aged 20–74 years old also support this correlation, pointing out the driving role of underlying conditions (HTN and diabetes) [228]. This increased risk seems to be related to several factors, including inflammation markers and vascular diseases, a sense of fatigue and lethargy during the day, and worsening of sleep fragmentation, which has been associated with atherosclerosis [242]. Moreover, long sleepers often have health issues, such as uncontrolled chronic diseases, OSAS or depression, or social discomfort due to low socioeconomic status, unemployment, or a low level of education [243,244], as shown in Figure 10.

A summary of mechanisms involved in the relationship between non-metabolic risk factor and CVDs has provided in Table 2.

## 5. Discussion

An evaluation of cardiovascular risk has evolved in the last few years. The optimistic expectations in managing traditional risk factors, such as HTN, hypercholesterolemia, hyperglycemia, and smoking, to reduce the burden of CVDs have been largely unmet. Clinicians and researchers have clearly realized that traditional risk factors may explain only part of the occurrence of acute events in the general population.

A risk factor is a factor associated with a greater probability of the onset of the disease. It must possess two fundamental characteristics: (1) constant (frequent) association and (2) plausible temporal sequence. An etiological or causal factor is a condition directly implicated in the determinism of the disease. It must meet the following requirements: biological plausibility, biological gradient of effects, strength of association, and specificity of the association. Starting from this statement, non-conventional risk factors are now emerging to better define cardiovascular risk profile. Several efforts have been made in exploring newer metabolic and non-metabolic risk factors and how they may affect cardiovascular outcome. The present article summarize the available evidence on these emerging factors and surrogates supporting the need for further researches to better address the controversial points.

On behalf of metabolic risk factors, homocysteine, UA and Vitamin D levels, gut microbiota status, Lp(a), and MS seem to be clearly linked to CVDs.

Although the role of homocysteine as a strong and independent cardiovascular risk factor is clear at present, conflicting data exist on the effect of the hyperhomocisteinamia lowering strategy and cardiovascular benefits. Observational studies and meta-analyses exploring the folic acid and vitamin B12 supplementation to reduce hyperhomocisteinemia seems to be beneficial in both primary and secondary prevention on the development of CAD and stroke and on the incidence of mortality from cardiovascular causes [245,246]. However, other prospective, randomized, case-controlled, and meta-analyses studies have shown no benefit of hyperhomocysteinemia treatment in the context of primary and secondary prevention of cardiovascular events (CAD, myocardial infarction, cardiovascular death, and all-cause mortality), except for a reduction in the risk of stroke, observed only in some of these meta-analyses [247,248,249,250,251,252,253,254]. Hence, current guidelines on cardiovascular prevention do not suggest serum homocysteine as standard practice in CVD prevention [3]. Better-designed clinical trials are needed to clarify the existing doubts on this regard.

Similarly, UA levels seem to offer a good picture of inflammatory status and coronary atherosclerosis in cardiovascular patients. Currently, the limit value for UA set by the Guidelines is <7 mg/dL in men and <6 mg/dL in women [255]. The URRAH observational study, which included 22,714 patients, defined the cutoff value > 5.6 mg/dL as associated with an increased risk of cardiovascular mortality [256]. Based on the available data correlating UA with the basic/clinical features of CVDs, the use of hypouricemic drugs even at an early age in patients with known CVD or other risk factors could represent a possible effective therapeutic strategy. However, current guidelines fail in defining a clear recommendation for this issue.

Vitamin D is another promising additional marker for cardiovascular evaluation, but the controversial findings from the clinical trial published to date have limited its predictive value. The primary challenge in investigating the relationship between vitamin D levels and CVD disease lies in distinguishing the cause–effect relationships from statistical correlations. While it is evident that vitamin D levels represent an unconventional cardiovascular risk factor, the existence of a direct causal relationship between vitamin D metabolism and CVD is still a subject of debate. The most controversial trial published to date is the VITAL study [257]. A total of 25,871 participants were enrolled to evaluate the effect of vitamin D supplementation on cardiovascular prevention [257]. However, only 15,787 vitamin D levels were available. Of these participants, only 12,7% (2005 subjects) were vitamin D deficient (with a value below 20 ng/dL), and 32,2% were insufficient (with a value between 20–30 ng/dL). Based on the available literature showing that cardiovascular risk increase for levels below 20 ng/dL [79,258], the number of deficient subjects in VITAL study seems to be too small for any conclusion. Taking into account the antithrombotic and anti-inflammatory properties reported in different experimental model, better-designed clinical trials are needed to finally clarify the role of vitamin D as a marker of CVD.

Gut microbiota is an organ with an important role in host’s metabolism due to several systemic effects. Intestinal dysbiosis, through the mechanisms previously described, represents a non-traditional cardiovascular risk factor [259]. A greater knowledge of microbe-microbe and microbe-host relationships could be the prerequisite for targeted strategies for microbiota modulation with the purpose to modify host’s immune-inflammatory and metabolic state in the desired direction.

Lp(a), with its distinctive composition, enigmatic functions, and substantial clinical implications, has ignited scientific interest and debate. As research progresses, a more profound comprehension of its pathophysiology could potentially unlock innovative diagnostic tools, therapeutic approaches, and preventive strategies, thereby enhancing our capacity to effectively manage and reduce risks linked to this intriguing lipid particle.

Several epidemiological and clinical studies have clearly shown the relationship between MS and CVD with an estimated risk up to 50–60% [125]. Taking into account that IR is the pathophysiological substrate of MS [127], its early detection by the available markers is of great importance. HOMA index is a reliable marker of IR [135]. It can be easily detected and should be considered by the current SCORE for CV definition.

Metabolic risk factors for CVDs are an evolving concept. Our understanding of their role in modulating cardiovascular pathways is increasing. A recent report has shown that even pre-menopausal breast fat density might predict cardiovascular outcome [260] because of its inflammatory, pro-apoptotic properties, and pleiotropic negative effects on the cardiovascular system [261]. This aspect is of great importance because despite the reported sex difference for incident and recurrent coronary events and all-cause mortality with lower risk in women [262], the presence of overweight and metabolic distress could cause major adverse cardiac events in women via over-inflammation [263] Thus, future researches should take into account most of these novel modulators to better define the metabolic CV risk of the general population.

Other non-metabolic risk factors with pathophysiological implications affect the cardiovascular system. Of these, OSA, air pollution, climate changes, and sleep duration may modify cardiovascular outcome; thus, they should be considered and quantify in defining cardiovascular risk.

OSA is clearly linked to any CVD, and because of its prevalence in general population, it should be added to the current score to define CV outcome. The chronic hypoxia and hypercapnia, induced by the mechanical collapse of the upper airways during sleep, leads to different functional and metabolic changes that, as discussed above, are responsible of the CVD pathogenesis. However, current studies have failed to show cardiovascular benefits from OSA treatment with CPAP [264]. These negative results seem to be related to the poor compliance of the patients to the treatment [264]. Hence, additional trials are needed to solve this issue.

On the contrary, the role of air pollution in CVD has been defined. Considering the evidence to date, the most recent guidelines of the European Society of Cardiology have identified air pollution as a major modifiable risk factor relevant to the prevention and management of CVD [3]. The APHEKOM project, conducted in 25 European cities, calculated that meeting the annual average PM_2.5_ values recommended by the WHO guidelines (annual average 10 mg/m^3^) would add up to 22 months of life expectancy at age 30, corresponding to a total of 19.000 delayed deaths [265]. A greater understanding of the mediators underlying the impact of air pollution on human health are needed to spur political forces to the implementation of targeted, effective, and enforceable legislation on global air pollution reduction in order to protect people at risk and reduce the effect on CVD.

Climate change is another well-defined non-conventional risk factor. The consequence of global warming is the exposure of the population to moderate to extremely hot temperatures and less exposure to the cold, with consequences for human health [266]. Several studies have suggested an increase in heat-related mortality. A reduction in risk is often considered a sign of adaptation, either as a result of a physiological acclimatization response to temperature changes (intrinsic adaptation) or through non-climatic factors that contribute to risk reduction (extrinsic adaptation), such as socioeconomic development or personal care [267]. Management of this risk factor should be part of a global strategy with green interventions able to reduce its impact in a close future.

Lastly, evaluation of sleep duration should become part of the medical examination since the available literature support its correlation with CVD [268]. Currently, the most important European and American associations for sleep and CVDs suggest a nocturnal sleep duration, preferably unfragmented, of about 7 h [269,270]. Daytime naps are discouraged, except for subjects who have a nocturnal sleep time below 6 h.

## 6. Conclusions

Management of CVD is evolving. Current evidence clearly indicates that beyond traditional risk factors, the medical community should start to consider different non-conventional factors and surrogates that may induce pathophysiological changes linked to CVD and outcome. The latest guidelines from international societies still fail to add these emerging factors and surrogates to the available SCORE for cardiovascular risk evaluation and better define the countries at risk taking into account their climate and air pollution status, too. Thus, a major effort should be made by researchers to generate a novel algorithm that by combining conventional and non-conventional risk factors might be more accurate for cardiovascular risk scoring.

## Figures and Tables

**Figure 1 biomedicines-11-02353-f001:**
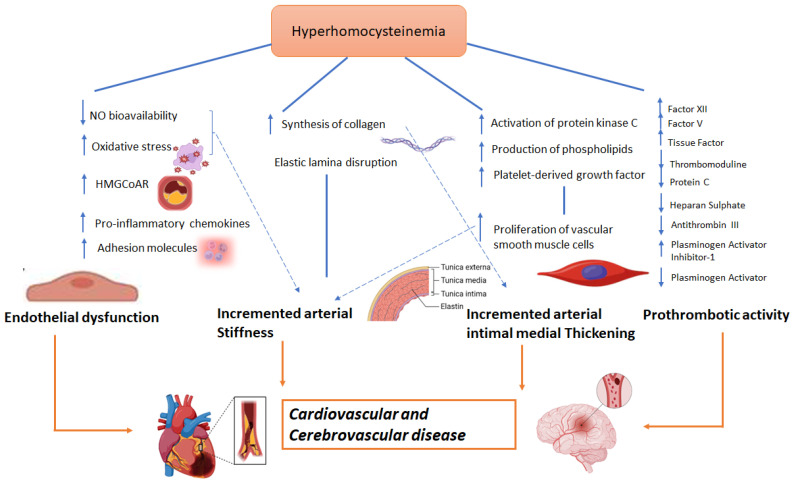
Possible role of homocysteine in CVD.

**Figure 2 biomedicines-11-02353-f002:**
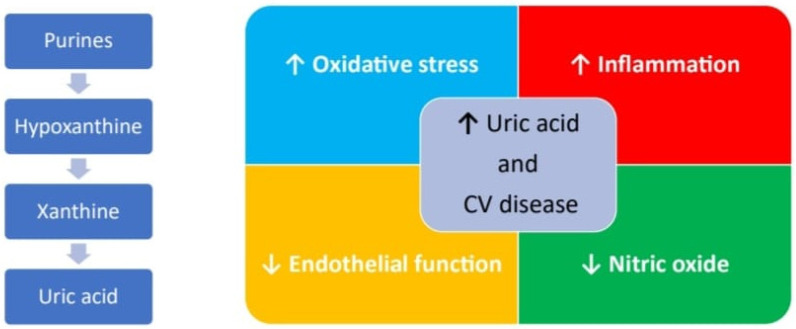
Major pathways UA related involved in pathogenesis of CVD.

**Figure 3 biomedicines-11-02353-f003:**
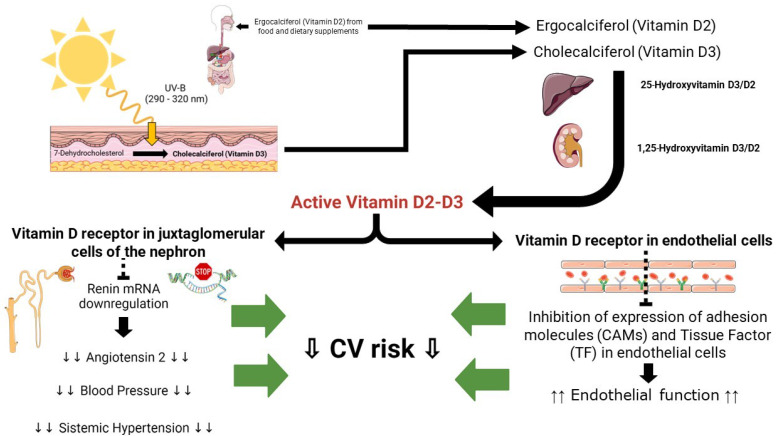
Putative cardiovascular pathways Vitamin D-related: see text for details.

**Figure 4 biomedicines-11-02353-f004:**
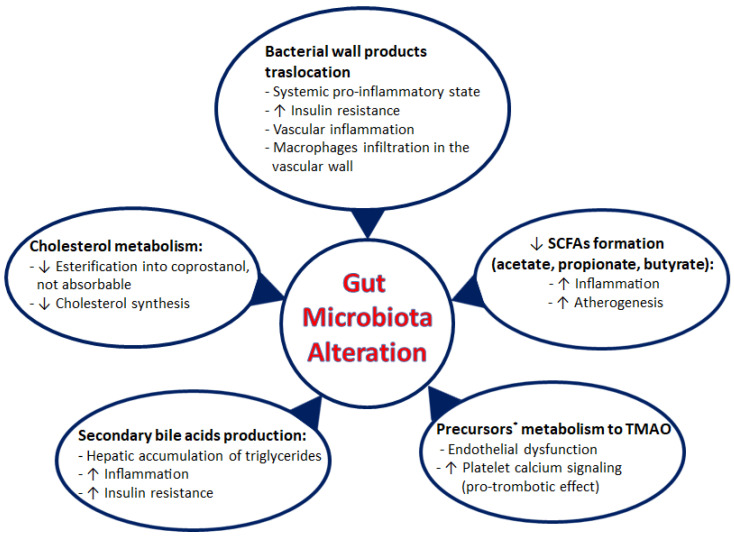
Summary of the main mechanisms by which gut microbiota, in condition of dysbiosis, influences the pathogenesis of atherosclerosis. * Precursors to TMAO: choline, L-carnitine and betaine. See text for details.

**Figure 5 biomedicines-11-02353-f005:**
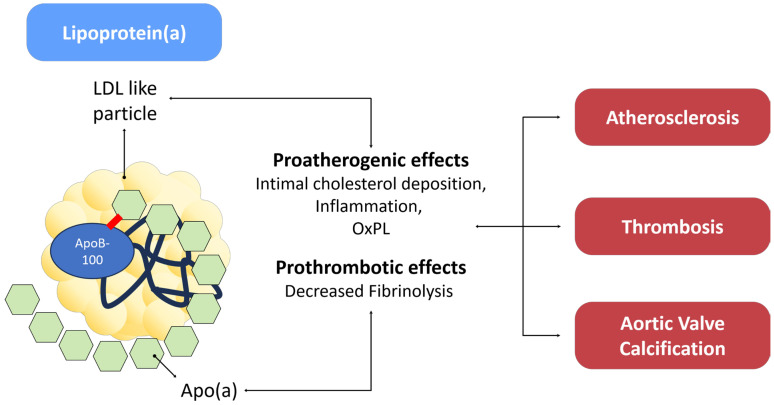
Lp(a) connection with CVD.

**Figure 6 biomedicines-11-02353-f006:**
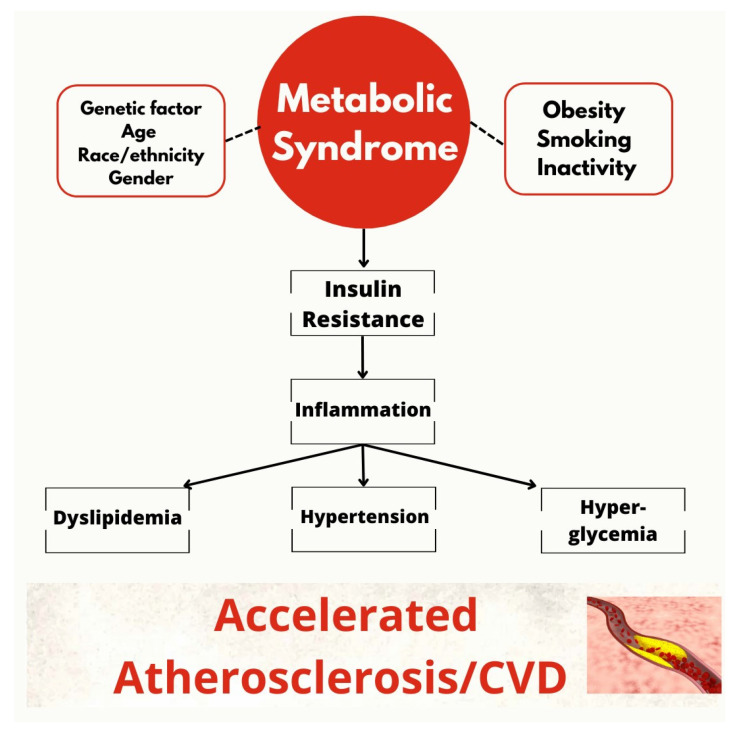
Schematic view of metabolic syndrome leading to CVD.

**Figure 7 biomedicines-11-02353-f007:**
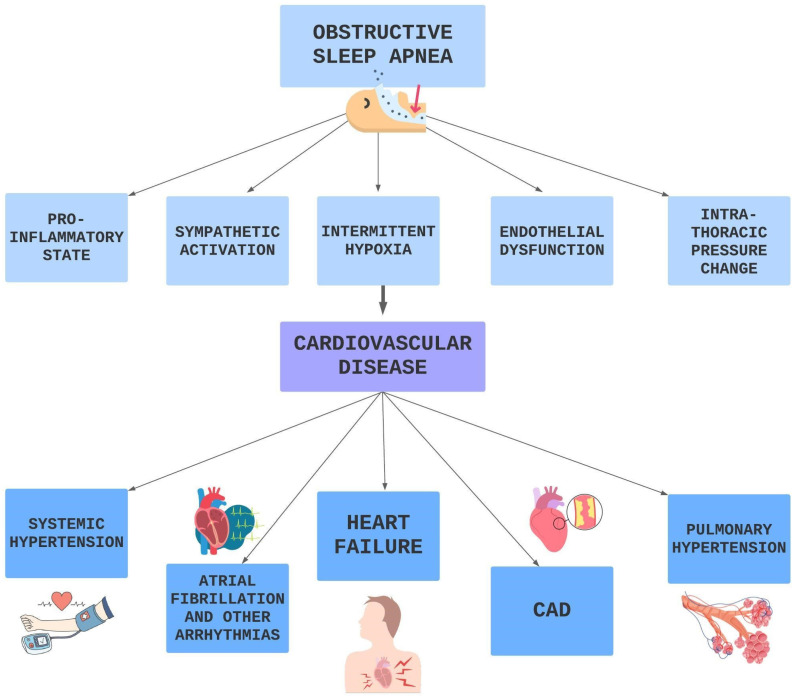
Pathophysiological pathways OSA related leading to CVD.

**Figure 8 biomedicines-11-02353-f008:**
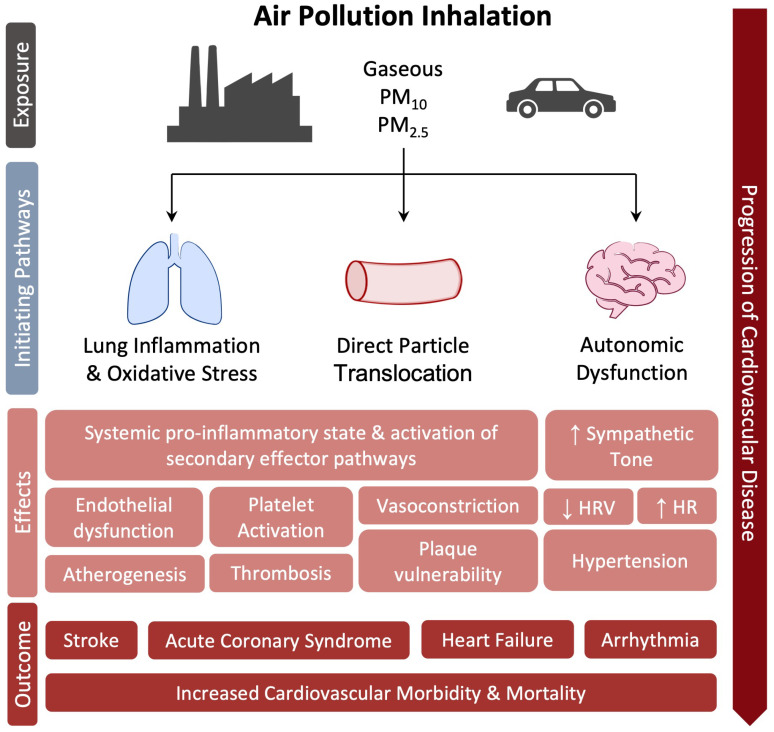
Molecular mechanisms linked air pollution to CVD.

**Figure 9 biomedicines-11-02353-f009:**
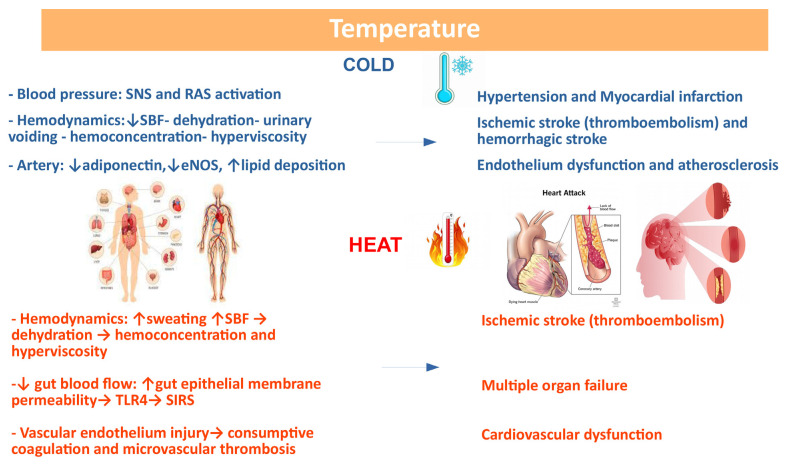
Correlation between climate changes and CVD: possible basic mechanisms. Several variables affect the response to temperature changes.

**Figure 10 biomedicines-11-02353-f010:**
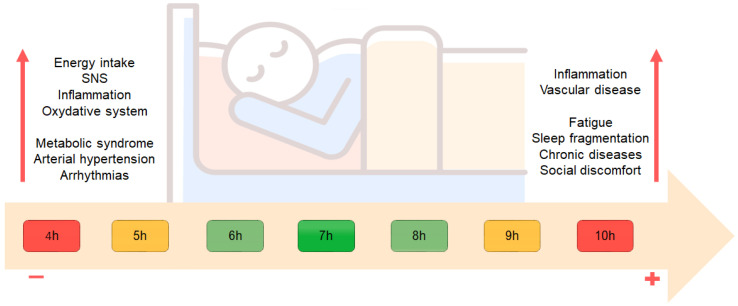
Association between Sleep duration and CVD.

**Table 1 biomedicines-11-02353-t001:** Metabolic risk factors.

Risk Factor	Observed Effects/Impact on Conventional CV Risk Factors	Mechanisms
Homocysteine	- CHD - Stroke - peripheral arterial disease - venous thromboembolism	- Endotelial disfunction: oxidative stress; Reduced NO bioavailability; increased expression of HMGCoAR; increased expression of CAM and pro-inflammatory interleukin (IL-8) - Incremented arterial stiffness: elastic lamina disruption, proliferation of smooth muscle cells and incremented synthesis of collagen - Incremented arterial intimal-medial thickening: proliferation of smooth muscle cells; incremented synthesis of collagen. - Prothrombotic state: activation of pro-coagulant factor: factor XII, factor V, TF; plasminogen activator inhibitor-1 and reduced activity or expression of anti-coagulant factor: thrombomodulin; protein C, Heparan Sulfate; Antithrombin III; plasminogen activator
Uric Acid	- Increased Oxidative stress - Reduced NO - Endothelial disfunction - Inflammation	The synthesis of uric acid determines the formation of ROS. ROS are responsible for the lipid oxidation and the reduction of the NO concentration which causes the loss of the normal endothelial function and induces a pro-inflammatory and pro-trombotic state
Vitamin D	- CVD risk reduction. - Effects on blood pressure.	- Reduced expression and activity of TF and CAMs on ECs induced by oxidized lipids or interleukin-6, possibly preserving endothelial function. - Vitamin D regulation of renin synthesis
Gut Microbiota Alteration	- Cholesterol reduction - Insulin resistance - Systemic pro-inflammatory state - Endothelial dysfunction - Pro-trombotic state	- Reduction of cholesterol synthesis and absorption - Bacterial wall product translocation - Reduced SCFAs formation - TMAO production
Lipoprotein(a)	- Atherosclerotic CVD	- Intimal cholesterol deposition - Inflammation - Lipid oxidation - Hemostasis impairment
Metabolic Syndrome	- Hyperglicemia - Hypertension - Dyslipidemia	- alteration of glucose transport by down-regulation of GLUT4, increased expression of protein tyrosine phosphatases which dephosphorylate and interrupt intracellular signaling - reduced NO production and hyperactivation of the sympathetic system - increased release of fatty acids from adipocytes; increased hepatic VLDL secretion and therefore hypertriglyceridemia; stimulation exchange of cholesterol esters from HDL

**Table 2 biomedicines-11-02353-t002:** Non-metabolic risk factors.

Risk Factor	Observed Effects/Impact on Conventional CV Risk Factors	Mechanisms
Obstructive sleep apnea syndrome	HTN, AF and other arrhythmias, HF, CAD, stroke, pulmonary hypertension, metabolic syndrome and diabetes	Hyperactivation of SNS; systemic oxidative stress; endothelial dysfunction; systemic inflammation; atherosclerosis; higher plasma leptin levels; glucose metabolism impairment and insulin resistance
Air Pollution	HTN, endothelial dysfunction, increased atherosclerotic plaque vulnerability and activation of prothrombotic and proarrhythmic state	Systemic oxidative stress & Inflammation, autonomic imbalance in favor of sympathetic tone
Air temperature	-Cold: HTN, atherosclerosis, stroke -Heat: stroke, multiple organ failure, cardiovascular dysfunction	-Cold: SNS and RAAS activation; lipid deposition; dehydration, urinary voiding and hemoconcentration -Heat: dehydration and hemoconcentration; gut epithelial membrane permeability and SIRS; vascular endothelium injury
Sleep duration	Increased CVD risk and HTN in both short and long sleep duration	Short: metabolic changes, hyperactivation of ANS, inflammation and oxidative system protein. Long: increased inflammation, vascular disease, atherosclerosis. Association to uncontrolled chronic diseases and social discomfort.

## Data Availability

The data presented in this study are available on request from the corresponding author.

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
