# Peer review of "Non-Conventional Risk Factors: “Fact” or “Fake” in Cardiovascular Disease Prevention?"

_biomedicines, 2023, doi:10.3390/biomedicines11092353_

Round 1
Reviewer 1 Report
The review of Cimmino et al. is of great interest as it exposes many of the main metabolic and non-metabolic cardiovascular risk factors. The organization of the paper makes the reading easier by dividing the different topics in an orderly manner.
The figures are of high quality and very explanatory.
I would suggest that authors include two additional paragraphs on two other risk factors that I believe are compatible with this review: 1. insulin resistance and its correlations with metabolic syndrome. 2 liprotein a and coronary risk.
In this perspective I suggest to insert and discuss the following references
1. doi: 10.3390/biomedicines9101356
2. DOI 10.1016/j.diabres.2021.1089593. DOI 10.3390/biom11121834
Minor editing of English language required
Author Response
We thank the reviewer for the time he/she spent to review our article and for the criticisms he/she raised. We felt they were appropriate and the revised version of our manuscript is improved because of that. To facilitate the readers, all changes are in red.
We have added two new parahraphs as suggested and enriched the references list as indicated
Reviewer 2 Report
I had read with great interest the nice and comprehensive review entitled “Non-conventional risk factors: “fact” or “fake” in cardiovascular diseases prevention?” which aim was to define emerging metabolic and non-metabolic factors and describe the potential mechanisms by which they might contribute to the development of CVD.
Risk factors for cardiovascular disease are habits, behaviors, or conditions that increase the risk of developing cardiovascular disease, but at the same time they should show that their treatment reduce the risk of developing cardiovascular disease.
On the other side, a biomarker is a parameter used to identify the likelihood of a clinical event, disease recurrence or progression in patients who have a disease or a medical condition.
In this scenario, risk factors and biomarkers are conceptually different and not interchangeable. Smoking habit, hypertension, diabetes, and dyslipidemia showed a powerful relationship to cardiovascular risk development and their treatment reduce that risk.
Other variables could be called markers or perhaps modulators. But to be considered as risk factors they should show treatment benefits.
This review showed elegantly the pathophysiological and epidemiological relationship between this non-conventional risk factors which does not imply a causal relationship.
Most of the treatments focus on the metabolic variables discussed by the authors had failed in demonstrating their clinical benefits. This topic should be comprehensively discuss by the authors as well as the limitations to consider them as risk factors vs. biomarkers.
In fact, most of the guidelines do not considered them as risk factors.
The case of the non-metabolic variables discussed by the authors is much more difficult to define. Obviously, the impact of air pollution and climate change and their treatment is almost impossible to establish. So it is very difficult to say if they are risk factors or markers. This limitations and ambiguities should also be comprehensively discuss by the authors.
In conclusion, although a pathophysiological and statistical association between some of these variables could be demonstrated, even independently of others, and cardiovascular disease, they are far away from being considered CV risk factors. In my humble opinion this should be the main message of the review.
In addition to the previous comments, I think that tables 1 and 2 are redundant with the graphics and text contents.
Author Response
We thank the reviewer for the time he/she spent to review our article and for the criticisms he/she raised. We felt they were appropriate and the revised version of our manuscript is improved because of that. To facilitate the readers, all changes are in red.
The reviewer has correctly pointed out the main difference between “true” and “potential” cardiovascular risk factors. We agree that it is very difficult to say if they are risk factors or surrogates. We have provided epidemiological data linked each of them with an increased risk for cardiovascular diseases. However, some of them are still not considered by the current guidelines. By summarizing in this article all the available evidence to date, it might be a stimulus for futher researches. Some changes could be encouraged even in the short term. A better definition of the areas with a huge impact of the climate changes and/or air pollution might be of help to rivisitate risk regions. Moreover, in the current SCORE system, the inclusion of OSA (yes/not), vitamin D (< 20ng/dL) and homocystein levels , spleep duration (<6hours or > 8 hours), could redefine CV risk. This may a matter for future research. We have added this sentence in the revised version of the manuscript
With the respect for reviewer suggestion, we would kept the tables since we believe they may be of help for the readers to have a summary of the metabolic changes induced by the risk factors and surrogates discussed
Reviewer 3 Report
This is an interesting and updated review on the novel (non-conventional) risks of cardiovascular diseases (CVD and related, such as CAD and others). The articles is divided in two different parts, the first one devoted to “novel” metabolic factors and the second one to “novel” non-metabolic factors. The first ones are based in clear scientific evidence, and the second ones are more controversial and difficult to prove, but the manuscript supplies recent meta-analysis strongly suggesting the involvement of these factors.
The following minor points should be addressed before definitive acceptance:
1) Line 96. The metabolic pathway related to homocysteine demand would be briefly discussed, or alternatively, the sentence about homocysteine accumulation in blood should be re-written. As far as I know, human metabolism has no demand of homocysteine. There is demand of methylation, and homocysteine is a subproduct that should be eliminated.
2) Concerning Figure 1.
Repair the Word “adesion” by adhesion.
There are two blood vessels (arteria or vein) at the left and in the middle bottom. Is possible to unified in just one?
The legend should include the word “possible” role in the light of the conflicting results obtained in the studies of some of the mentioned biochemical parameters. This point is appropriately discussed in the text (i.e. line 164).
Is protein C the same that C-reactive protein? If so, please uniform the terminology throughout the manuscript.
3) Replace TNF by TNFalpha throughout the manuscript. Other Greek letters should be checked (i.e. kappa in the NFkB transcription factor)
4) Figure 4: Is only choline transforming into TMAO, what about carnitine and betaine?
5) Replace Interleuchine by interleukin.
6) Re-write legend of Figure 5.
7) Some paragraphs should be justified.
8) The final risks (mainly seep duration) are written a little bit verbose... this is likely because some points are difficult to be proven, and controversial. I.e., cold and heat give place to the same symptoms (dehydration, hemoconcentration, hyperviscosity). This should be clarified ... ¿?. On the other hand, it is stated that males are more vulnerable to cold than females (line 549). The opposite has also been published. Please, discuss this point of give more references.
9) Line 620, replace die by day.
10) The use and definition of the abbreviations use should be uniformed. There is a list at the end of the text (this is right, I recommend the maintenance), but some abbreviations are not included in the list and they should be. Other abbreviations are defined throughout the text or at the footnote of Table 1, sometimes in a repetitive way, at different paragraphs. It seems that metabolic and-non metabolic factors have been written by different authors. An integrative work for unified style would be great.
Minot hypos should be corrected
Author Response
This is an interesting and updated review on the novel (non-conventional) risks of cardiovascular diseases (CVD and related, such as CAD and others). The articles is divided in two different parts, the first one devoted to “novel” metabolic factors and the second one to “novel” non-metabolic factors. The first ones are based in clear scientific evidence, and the second ones are more controversial and difficult to prove, but the manuscript supplies recent meta-analysis strongly suggesting the involvement of these factors.
We thank the reviewer for the time he/she spent to review our article and for the criticisms he/she raised. We felt they were appropriate and the revised version of our manuscript is improved because of that. To facilitate the readers, all changes are in red.
The following minor points should be addressed before definitive acceptance:
- Line 96. The metabolic pathway related to homocysteine demand would be briefly discussed, or alternatively, the sentence about homocysteine accumulation in blood should be re-written. As far as I know, human metabolism has no demand of homocysteine. There is demand of methylation, and homocysteine is a subproduct that should be eliminated.
ANSWER. We apologize for the misunderstand. The sentence has been re-written (page 3, line 95)
2) Concerning Figure 1.
Repair the Word “adesion” by adhesion.
There are two blood vessels (arteria or vein) at the left and in the middle bottom. Is possible to unified in just one?
The legend should include the word “possible” role in the light of the conflicting results obtained in the studies of some of the mentioned biochemical parameters. This point is appropriately discussed in the text (i.e. line 164).
Is protein C the same that C-reactive protein? If so, please uniform the terminology throughout the manuscript.
ANSWER. We apologize for the typos. Figure 1 and legend have been changed according reviewer’s suggestions. Protein C is not CRP but the glycoprotein synthesized in the liver with Protein S, which role is to maintain the physiologic function of coagulation within the body.
3) Replace TNF by TNFalpha throughout the manuscript. Other Greek letters should be checked (i.e. kappa in the NFkB transcription factor)
ANSWER. We thank the reviewer for his/her suggestion. The Greek letters have been checked as suggested
4) Figure 4: Is only choline transforming into TMAO, what about carnitine and betaine?
ANSWER. We thank the reviewer for his/her suggestion. We have replaced “Choline” with “Precursors” indicating in the legend choline, carnitine and betaine as already indicated in the main text (page 8, line 321). A updated reference has been also added (10.1111/1750-3841.15186)
5) Replace Interleuchine by interleukin.
ANSWER. We apologize for the typos. The word has been replaced as indicated
6) Re-write legend of Figure 5.
ANSWER. Legend of Figure 5 (that is now Figure 7) has been re-write as suggested
7) Some paragraphs should be justified.
ANSWER. We apologize for the non homogeneous format. All the paragraphs have been justified as indicated
8) The final risks (mainly seep duration) are written a little bit verbose... this is likely because some points are difficult to be proven, and controversial. I.e., cold and heat give place to the same symptoms (dehydration, hemoconcentration, hyperviscosity). This should be clarified ... ¿?. On the other hand, it is stated that males are more vulnerable to cold than females (line 549). The opposite has also been published. Please, discuss this point of give more references.
ANSWER. We thank the reviewer to point this out. We agree with his/her evaluation and to avoid any counfounding factor, we have revisited the two paragraphs. Very recent references have been also added.
9) Line 620, replace die by day.
ANSWER. We apologize for the typos. The word has been replaced as indicated
10) The use and definition of the abbreviations use should be uniformed. There is a list at the end of the text (this is right, I recommend the maintenance), but some abbreviations are not included in the list and they should be. Other abbreviations are defined throughout the text or at the footnote of Table 1, sometimes in a repetitive way, at different paragraphs. It seems that metabolic and-non metabolic factors have been written by different authors. An integrative work for unified style would be great.
ANSWER. We apologize for the non-uniformed style. Abbreviations have been defined at the first time of appearance and listed at the end. We have added the missed abbraviations and remove them from the footnote of the tables as suggested
Comments on the Quality of English Language:
Minot hypos should be corrected
ANSWER. Minor typos have been corrected
Round 2
Reviewer 2 Report
Unfortunately, the author´s review is not satisfactory
Author Response
Thank you for your comments!